# Unlocking the Potential of Public Datasets: Wastewater-Based Epidemiological Forecasting During COVID-19

### Zhicheng Zhang
zczhang@cmu.edu
Carnegie Mellon University
Pittsburgh, PA, USA

### Sonja Neumeister
sneumeister@ucdavis.edu
University of California Davis
Davis, CA, USA

### Angel Desai
andesai@ucdavis.edu
University of California Davis
Sacramento, CA, USA

### Maimuna Shahnaz Majumder
maimuna.majumder@childrens.harvard.edu
Boston Children's Hospital, Harvard
Medical School
Boston, MA, USA

### Fei Fang
feifang@cmu.edu
Carnegie Mellon University
Pittsburgh, PA, USA

## ABSTRACT

The COVID-19 pandemic has emphasized the necessity for effective tools to monitor and predict epidemiological trends. Traditional approaches to disease surveillance possess certain limitations, leading to the emergence of wastewater-based epidemiology (WBE) as a complementary approach. WBE has demonstrated a strong correlation with traditional epidemiological indicators (e.g., number of clinical cases and hospitalization), which makes it a valuable asset in informing public health decision-making processes. Despite the promising prospects of WBE, it faces two main challenges, restricted data accessibility and high intrinsic noise and distribution shift in the data. In this study, we examine the feasibility of utilizing exclusively two publicly available data, specifically aggregated wastewater data and reported case counts, for epidemiological forecasting in the COVID-19 pandemic. We incorporate a variety of statistical and machine learning models in an attempt to address the inherent volatility and bias of the data. We further introduce the usage of the segmentation method during the evaluation phase as a better evaluation metric. Our empirical results show that, even with limited data, performing epidemiological forecasting is possible, and its performance is comparable with methods that use more diverse data sources, suggesting its potential for broader health applications. Additionally, we utilize the insights from results on the length of the forecasting horizon to provide practical guidelines regarding real-world prediction.

## KEYWORDS

COVID-19, Disease Surveillance, Wastewater-Based Epidemiology, Time-Series Forecasting

**ACM Reference Format:**

Zhicheng Zhang, Sonja Neumeister, Angel Desai, Maimuna Shahnaz Majumder, and Fei Fang . 2023. Unlocking the Potential of Public Datasets:

Wastewater-Based Epidemiological Forecasting During COVID-19. In *epiDAMIK 2023: 6th epiDAMIK ACM SIGKDD International Workshop on Epidemiology meets Data Mining and Knowledge Discovery, August 7, 2023, Long Beach, CA, USA.* ACM, New York, NY, USA, 8 pages.

## 1 INTRODUCTION

The COVID-19 pandemic has emphasized the importance of reliable tools for monitoring and forecasting epidemiological trends. Traditional disease surveillance approaches, based on clinical data, have limitations in both timeliness and coverage. Wastewater-based epidemiology (WBE) has thus emerged as a complementary approach to track the spread of infectious diseases in communities [8]. WBE has demonstrated significant potential in the monitoring and forecasting of epidemics, particularly during the COVID-19 pandemic. Several studies have utilized wastewater data to forecast clinical cases, hospitalizations, and ICU admissions, as well as to evaluate the effectiveness of governmental policies in containing COVID-19 transmission [10, 12, 13, 27]. Studies have found a strong link between data from wastewater surveillance and disease indicators. This link can help make better health decisions, use resources wisely, and put interventions in place quickly.

However, despite the promising results of WBE, there are two main challenges that need to be addressed for broader practical applications, which haven't been thoroughly explored in the existing literature. First, current approaches in using WBE mainly rely on small-scale, privately collected data, such as those from university campuses [36], or inaccessible private-sector wastewater data [10, 12]. Often, methods supplement wastewater data with additional data sources, including Community Vulnerability Index (CCVI) and vaccination records [13]. In a broader context, the sharing of wastewater data is restricted, and its coverage is geographically skewed towards economically developed areas that have a greater number of wastewater monitoring facilities [18, 23]. Second, the real-world epidemiological data is inherently noisy due to various factors such as sampling errors and challenges in attributing causes [24]. This issue is further exacerbated during global pandemics like COVID-19, where the temporal correlations within the data can drastically shift over the course of the pandemic, undermining the accuracy of predictions. Such drastic shifts can occur when a new variant emerges and rapidly becomes dominant or when vaccination rates significantly increase, both of which

cause distinct changes in epidemiological trends. These shifts underscore the need for robust forecasting models capable of adapting to evolving pandemic dynamics.

In this study, we focus on two publicly available datasets: aggregated wastewater data and reported case counts, both at the country level. This selection of datasets is driven by the ready accessibility and reliability of these data sources: wastewater data is regularly published not only by the CDC's National Wastewater Surveillance System (NWSS) but also by other agencies adhering to CDC protocols, while case count numbers are widely reported. This widespread adoption of consistent data-gathering protocols ensures the broad availability and comparability of these datasets. It also aims to alleviate volatility and mitigate biases inherent in smaller or less developed regions. The COVID-19 pandemic's landscape has been constantly changing, influencing how we assess its spread and impact. Initially, the case count data, encompassing both severe and mild cases, offered valuable insight into the pandemic's trajectory. This metric was particularly comprehensive during periods of widespread testing and reporting. However, as the pandemic has progressed, testing methods and reporting practices have evolved, with an increase in home testing and a decrease in reports to governmental agencies. While these changes present challenges, case count still servers as a strong signal of disease prevalence. Our core objective here is to investigate the feasibility of using only these two publicly available data sources, case counts and wastewater data, for epidemiological forecasting.

To evaluate this feasibility, we model the problem as a time-series forecasting problem characterized by significant distribution shifts in the data over time. We employ data preprocessing techniques to manage misaligned time-series data and introduce a segmentation algorithm during the evaluation phase to account for temporal shifts. This segmentation method enhances evaluation accuracy by ensuring that the test data spans only one wave so that the test error would no longer be masked by the results in other waves, and we empirically evaluate it to be a better evaluation criterion. To balance interpretability, simplicity, and prediction accuracy, we implement a variety of statistical and machine learning models, including linear regression, ARIMAX, Gaussian Process Regression, multilayer perceptron (MLP), and Long Short-Term Memory (LSTM) networks. The diversity of these modeling techniques enables us to compare the efficiency of simpler models with their more complex, deep-learning counterparts. Finally, our analysis shows that by only using aggregated wastewater data and reported case counts, we can achieve comparable performance with a random-forest model trained on diverse data sources, including CCVI indexes, and vaccination records in [13]. We further empirically demonstrate that the segmentation method provides a more accurate evaluation, particularly during volatile periods such as the case count peak in early 2022. Based on the empirical results on the effect of forecasting horizon of different lengths, we provide a practical recommendation for selecting the forecasting horizon in order to optimize the balance between reaction time and prediction accuracy.

## 2 RELATED WORK

*Wastewater-based epidemiology.* Wastewater-based epidemiology (WBE) has become an important tool for monitoring and forecasting epidemiological trends over the past two decades [8]. During the recent outbreak of COVID-19 [6], wastewater data was used to forecast clinical cases, hospitalizations, and ICU admissions, as well as to evaluate the effectiveness of governmental policies [10, 12, 12, 13, 27]. Galani et al. [10], Kaplan et al. [12], Stephens et al. [27] measured the wastewater for a number of monitoring sites and empirically demonstrated a strong correlation between hospitalizations and wastewater surveillance data using regression models. Kaplan et al. [12] used wastewater data to estimate reproductive numbers. Li et al. [13] used data from 100 USA counties to predict hospital and ICU admission numbers using random forest models.

However, despite its effectiveness in predicting epidemiological trends, wastewater data were not widely shared with the public or accessible to researchers, making it infeasible to perform additional analyses [18]. Current works often rely on small-scale, privately collected dataset [36], or supplement the dataset with other diverse sources of data, like vaccination records and CCVI indexes [13]. In addition, the coverage of wastewater data is severely biased toward economically more developed geographic regions with more wastewater monitoring facilities [18, 23]. In an attempt to address these challenges, our approach differs from previous work in that we aim to assess the promise of using exclusively two publicly available data sources: aggregated wastewater data and the reported case count data that are easily accessible to the public for epidemiological forecasting. Specifically, we focus on data within the United States while averaging it across the country to minimize bias in wastewater data from smaller or less-developed counties and states.

*Time-series forecasting.* Time series forecasting has been a longstanding problem in the fields of statistics and machine learning, attracting significant research attention. Classical methods [3, 16] provide a comprehensive understanding of time series analysis and forecasting and offer both theoretical insights and statistical guarantees. The advent of deep learning-based methods, particularly recurrent networks, has substantially improved the ability to capture temporal correlations in training data, as demonstrated by works including recurrent neural networks (RNNs) [22] and long short-term memory (LSTM) networks [11]. In recent years, long-term series forecasting (LSTF) research has focused on transformer-based models [30] due to their remarkable success in various application domains, such as natural language processing (NLP) [20] and computer vision (CV) [15]. Transformer-based LSTF models [14, 32, 34, 37, 38] have demonstrated impressive forecasting performance while also prioritizing prediction efficiency. However, recent criticism by Zeng et al. [35] suggests that the self-attention mechanism in transformers inevitably leads to temporal information loss, and their empirical results indicate that these models may not even outperform simple one-layer linear models in certain experiments.

In the domain of time series forecasting with scarce data, deep learning models frequently adopt less complicated architectures to enhance model performance. Tsaur [29] employed fuzzy grey regression models, while Abdulmajeed et al. [1] utilized an ensemble

of several auto-regressive models to improve accuracy and robustness in predicting COVID-19 cases in Nigeria. Informed by these insights, our approach emphasizes the use of simpler and more interpretable models when working with limited wastewater and case count data aggregated across the country. Specifically, we employed linear regression models, ARIMAX models, and Gaussian process regression models with a combination of kernels to address the problem of noise in the data. Additionally, we conducted a comparative analysis with deep learning models, including multi-layer perceptron (MLP) and LSTM models, to evaluate the effectiveness of our chosen methodology in the context of limited data.

## 3  PRELIMINARIES

*Time-series forecasting.* The primary objective of time-series forecasting [19, 25] is to make accurate predictions of future values in a sequence, utilizing historical observations as a basis. Consider a set of observed data points $\mathbf{x}_1, \ldots, \mathbf{x}_t$, where $\mathbf{x}_i \in \mathcal{X}$, the aim is to forecast the corresponding labels $\mathbf{y}_1, \ldots, \mathbf{y}_t$ for each timestep, ranging from 1 to $t$, with $\mathbf{y}_i \in \mathcal{Y}$. Let $h$ represent the look-back window size; when predicting the label $\mathbf{y}_i$, the prediction model can take as input $\mathcal{H} = \{\mathbf{x}_{i-h+1}, \ldots, \mathbf{x}_i\}$ or $\mathcal{H} = \{\mathbf{x}_{i-h+1}, \ldots, \mathbf{x}_i, \mathbf{y}_{i-h+1}, \ldots, \mathbf{y}_{i-1}\}$. This constraint ensures that predictions rely solely on information available within the specified historical context.

*Wastewater-based Epidemiology.* Wastewater-based epidemiology (WBE) is an approach to public health surveillance that leverages the detection of biological or chemical markers present in sewage to reflect the health status of a region [21]. In the case of COVID-19, the wastewater data measures genetic fragments of SARS-CoV-2 virus excreted in stool, specifically targeting the N1 and N2 regions of the nucleocapsid gene, to determine COVID-19 concentrations.

## 4  METHOD

In this section, we detail our data preprocessing steps, modeling techniques, and evaluation methods. Our focus of the training method lies in aligning misaligned time-series data, computing input embeddings, and employing models that strike a balance between simplicity, interpretability, and predictive accuracy. We also introduce a wave-based segmentation approach for evaluation, arguing its effectiveness as a more accurate metric and discussing its calibration using expert-identified waves.

### 4.1  Data Processing

To ensure the quality and consistency of the data used for training and evaluation, we first address the challenge of misaligned time series data and then segment the data into waves based on the observed distribution shifts. These preprocessing steps aim to improve the model's reliability and adaptability to changes in the underlying data distribution over time.

*4.1.1  Handling Misaligned Time-Series Data.* Dealing with inconsistent time intervals or irregular timestamps in time-series forecasting is a common challenge. In our study, the primary issue arises from weekly updates of wastewater data ($x_i$) and the daily updates of case count data ($y_i$). There are two main strategies to

address this: removing data points without corresponding labels or utilizing all available data, for instance, through interpolation [31].

Our approach is to associate each element $x_t$ in the wastewater dataset $\mathcal{X}$ with all elements that fall within the interval between two successive wastewater data updates. Specifically, for each $x_t$ in the dataset $\mathcal{X}$, we define:

$$x_t = \{x_t\} \cup \{y_i \mid T_{x_{t-1}} < T_{y_i} < T_{y_t}\} \tag{1}$$

where $T_x$ denotes the timestamp of the event $x$, and $y_t$ is treated as the ground truth label. The augmented $x_t$ now includes the wastewater data point at time $t$ and all case count data points whose timestamps $T_{y_i}$ are strictly greater than the timestamp $T_{x_{t-1}}$ of the preceding wastewater data point and strictly less than the timestamp $T_{x_t}$ of the current wastewater data point. The reason behind this decision is to maximize data utilization. However, it may not always reflect real-world scenarios, where all data might not be up-to-date, or future trends a few days from now need to be predicted. We empirically evaluate the impact of such delays when doing forecasting in Section 5.5.

*4.1.2  Embedding of input data.* As shown in Figure 1, there exists a lead-lag relationship [4, 13] between the wastewater data and the case count data. Specifically, signals in the wastewater data often precede signals in the case count data by a span of several days or weeks. To accommodate this time-shifted relationship, we implement a sliding window approach for both the wastewater and case count data inputs.

Formally, for a selected time point $i$, and a window size $h_w$ for wastewater data and $h_c$ for case count data, we generate input sequences $X_i^{wastewater}$ and $X_i^{casecount}$ respectively, as:

$$\begin{aligned} X_i^{wastewater} &= [w_{i-h_w}, ..., w_{i-l_w}] \\ X_i^{casecount} &= [c_{i-h_c}, ..., c_{i-l_c}], \end{aligned} \tag{2}$$

where $w_j$ denotes the wastewater data and $c_j$ denotes the case count data at time $j$. $l_c$ and $l_w$ are used to simulate the information available at the time of prediction in the real-world. $l_w = l_c = 1$ means that the prediction model is given all the data up-to-date.

To maintain scale consistency across all data points, we normalize the case count data using a min-max scaler, deriving the scaling parameters from historical data. This process ensures the data maintains its inherent trend and distribution characteristics while being compatible with the model input, especially the deep learning models.

### 4.2  Modeling Techniques for Time-series Data

In the context of limited data, the ideal model to capture temporal correlations should balance simplicity, interpretability, and a lower parameter count. More complex models, while potentially improving performance, might overfit the data and compromise interpretability and deployability. Therefore, in this study, our emphasis is on methodologies that ensure adequate predictive accuracy while maintaining computational feasibility and transparency in interpreting data patterns.

(1) Linear Regression Model [17]: Used as a benchmark, this simple model provides a baseline for performance comparison.

(2) ARIMAX Model [2]: Serving as a robust statistical model, ARIMAX extends the traditional ARIMA model by incorporating exogenous inputs, which helps in modeling complex temporal structures in the presence of influential external factors, which suits our dataset with a lead-lag relationship.

(3) Gaussian Process Regression (GPR) Model: This model leverages a custom kernel for handling non-linear relationships and noisy data. Our kernel construction, formulated as below, involves a multiplicative interaction of Constant and RBF kernels, along with an additive incorporation of a White kernel for noise management and a Matern kernel for smoothness.

(4) Multi-layer perceptron (MLP): A widely employed neural network for regression problems, our implementation features two hidden layers with 128 units each and ReLU as the activation function.

(5) Long Short-Term Memory (LSTM) model [11]: As a type of recurrent neural network, LSTMs are capable of capturing temporal dependencies in data, making them well-suited for time series forecasting tasks. LSTMs can learn to filter out noise by selectively retaining valuable information through gating mechanisms. To mitigate overfitting, we incorporate a dropout [26] rate of 0.5 after each layer in the model and added an $L_2$ regularization.

## 4.3 Wave-based Segmentation

One important observation for pandemic-related data is the dynamic nature of the underlying distribution over time. This variability can be attributed to several factors, including the emergence of different viral variants [5], changes in vaccination status among the population [7], and the implementation of varied government policies [33]. The presence of these distribution shifts significantly complicates the prediction process. To address this issue, we propose splitting the data into waves, where each wave is assumed to have a relatively stable distribution. We employ Binary Change Point Detection [9] for identifying time-series data change points, chosen for its multiple change point detection, no predetermined change point requirement, and computationally efficient $O(Cn\log n)$ complexity.

*4.3.1 Hyperparameter Calibration.* Once the waves are identified, we calibrate the model's hyperparameters, including the cost function, penalty term, and minimal distance between two change points, to fit the waves recognized by domain experts. We formulate a scoring function and select the optimal hyperparameters on the validation data. Given a set of detected change points $CP = \{cp_1, cp_2, \ldots, cp_n\}$ and a set of expert-identified waves $W = \{w_1, w_2, \ldots, w_m\}$, we define a score function as

$$S(CP, W, \alpha, \beta) = \sum_{i=1}^{m} \exp(-\alpha d(w_i, CP)) \\ - \beta |n - m|, \qquad (3)$$

where $\alpha$ is the decay factor for the impact of the distance between the detected change points and the actual waves, $\beta$ is the penalty coefficient that penalizes the absolute difference between the number of detected waves and the number of actual waves, $d(w_i, CP)$ denotes the closest distance between wave $w_i$ and the set of detected

change points in $CP$. The objective is to find hyperparameters that minimize this score:

$$CP^{\star} = \arg\min_{\alpha, \beta} S(CP, W, \alpha, \beta). \qquad (4)$$

Minimizing this metric allows us to select the hyperparameters that optimally align the detected change points with the expert-identified waves while balancing proximity and the penalty for the difference in the number of change points and waves.

*4.3.2 Evaluation using Wave-based Segmentation.* Our approach leverages wave-based segmentation for evaluation. Once we separate our dataset $D$ into training, $D_{\text{train}}$, and testing sets, $D_{\text{test}}$, we restrict the test data to have just one segment. Mathematically, if $S_{\text{test}}$ represents all segments in $D_{\text{test}}$, we would ensure that $|S_{\text{test}}| = 1$.

This methodology mirrors real-world conditions more accurately, as predicting data of new waves often requires substantial additional information. We avoid using wave-based segmentation in training due to potential data leakage issues, as it commonly uses global data to determine segmentation, which could inadvertently affect the results.

## 5 EXPERIMENTS

In this section, we outline the experimental setup, including data visualization and segmentation results, and present the empirical results obtained by evaluating the five models for the task of predicting case counts.

## 5.1 Experimental Setup

Our experiments exclusively use publicly available data, namely wastewater data[1] and case count data[2], count and death which are originally aggregated at the county or state level and therefore, pose inherent challenges due to their noisy nature. The case count data serve as ground truth for our prediction task. Owing to variability in the collection of country/state-level data, we aggregate all data at the national level and utilize the nationwide average for our analysis. Composed of wastewater data and case count data, our dataset spans from January 15, 2020, to February 15, 2023. Wastewater data is reported on a weekly basis (162 data points), while case count data are collected daily (1128 data points). For all the experiments, we report the mean and standard deviation of 6 runs.

To better understand the correlation between wastewater data and the case counts, we visualize the trends in the data in Figure 1. We aggregate the data at the national level due to the high variability and statistical noise inherent in the state-wise data, as evidenced in Figure 1(b). As shown in Figure 1 with the shifted wastewater curve, a strong association exists between the trend of virus concentration levels in wastewater and that of the number of cases, with wastewater data trends slightly preceding that of case counts. However, it is important to underscore that despite the exhibited association between the two trends, the relationship between their absolute numbers is not straightforward.

---

[1]https://github.com/biobotanalytics/covid19-wastewater-data
[2]https://usafacts.org/visualizations/coronavirus-covid-19-spread-map/

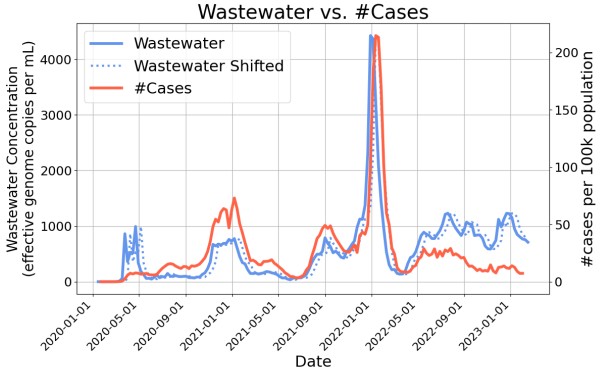

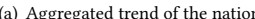

(a) Aggregated trend of the nation

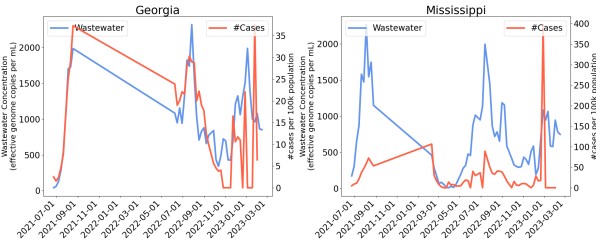

(b) Trend in Georgia and Mississippi

**Figure 1: Temporal Correlation between Wastewater Viral Concentrations and Case Counts per 100k population. The x-axis shows the dates ranging from 2020-01-15 to 2023-02-15, and the y-axis denotes the values of the viral wastewater concentrations and the number of cases per 100k population. Subfigure (a) describes the aggregated trend of the nation, and (b) describes two randomly picked states of Georgia and Mississippi.**

## 5.2 Visualization of Segmentation Result

After calibrating the hyperparameters on the expert-identified waves from March 2020 to February 2022 [28], we use the Binary Change Point algorithm [9] to detect the change points in the wastewater virus concentration level data. In our case, the expert data segmentation consists of five points, forming six distinct waves. As a result, we opted to include all of these points for the calculation of the score function during the calibration process. Figure 2 demonstrates that the detected change points closely align with the expert-identified waves and that our method can accurately detect change points even in areas not covered by the expert data segmentation.

## 5.3 Evaluation across Varied End Dates

To assess the accuracy of our models, we evaluate their performance throughout the course of the pandemic. Figure 3 represents the Normalized Root Mean Square Error (NRMSE) of each model over the different end dates, allowing for a comparative analysis of model consistency and adaptability across time. We compare our results

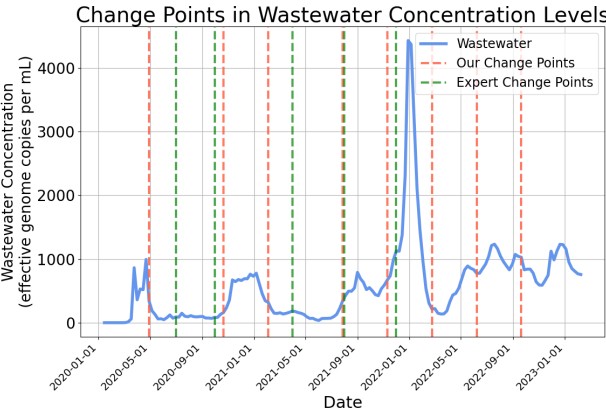

**Figure 2: Segmentation results using Binary Change Point Detection. The green dotted lines represent expert-identified change points, while the red dotted lines indicate our detected change points. The x-axis denotes the days passed since 2020-01-15, and the y-axis shows the viral wastewater concentration level. Our model's detected change points exhibit close correspondence with expert-identified points.**

with a random forest model developed by Li et al. [13]. Their model was trained on diverse data, including hospitalization and ICU admission records, CCVI indexes, and vaccination records, among others. Notably, their work does not clearly delineate the date range for the test data—a factor that could significantly impact the model's accuracy.

Figure 3 shows that the models perform relatively poorly in the early stages of the pandemic but improve significantly in the later stages, even during a sudden peak in early 2022. In the later stages of the pandemic (after July 2021), as shown in Figure 3, all five models reach performance on par with the baseline model, indicating an NRMSE below 1.0. This suggests that, on average, the model's prediction error is less than the standard deviation of the observed data, which is over 200 cases during the peak. The performance at the early stages is worse, possibly due to the lack of sufficient data to learn the inherent temporal correlation.

## 5.4 Impact of Segmentation on Evaluation

In addition to evaluating the performance on different dates, we also conduct an experiment to understand how wave segmentation impacts the evaluation of our models. Figure 4 shows model performance with and without segmentation for the models. Performance differences are more noticeable during peak periods, likely due to rapid trend shifts that make the prediction task difficult.

We remark that this experiment highlights the importance of segmentation in this task of predicting case counts, particularly during volatile periods. The omission of this segmentation method, as is the case in [13], could lead to inaccuracies in the Normalized Root Mean Square Error (NRMSE) as multiple waves in the test data may mask inaccuracies with one particular wave. Therefore, we present the results with the segmentation evaluation method for all subsequent experiments. It is also worth noting that these

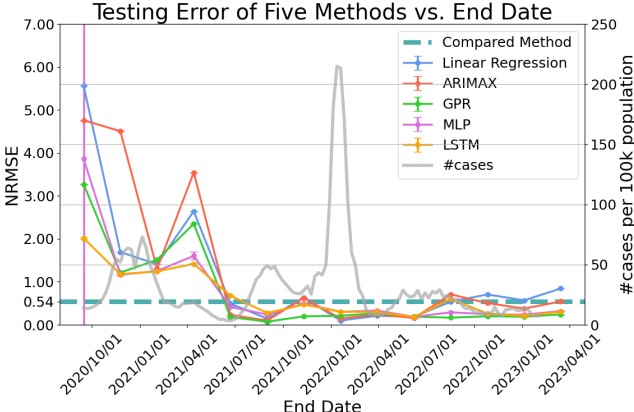

**Figure 3: Performance comparison of models across end dates. The x-axis denotes the end date of the test period, while the y-axis represents the normalized root mean square error (NRMSE) of the prediction for the number of cases. The grey curve denotes the actual number of cases. The dotted line denotes the reported performance of the model in [13].**

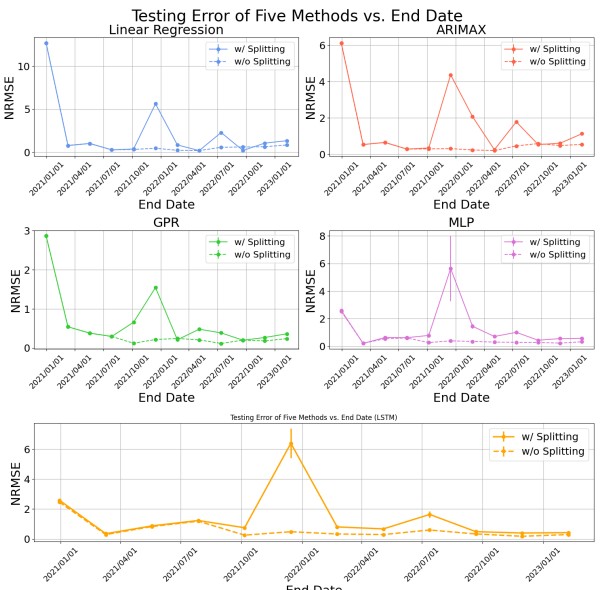

**Figure 4: Prediction accuracy comparison for each model with and without segmentation. The x-axis is the end date, and the y-axis is the normalized root mean square error (NRMSE) of the prediction for the number of cases. The dotted lines denote evaluation results with segmentation performed, and the solid lines denote evaluation without segmentation.**

results are based on the assumption of perfect up-to-date knowledge. Results based on more relaxed assumptions are discussed in the following subsection.

## 5.5 Prediction Accuracy across Varied Forecasting Horizon

We further examine our models' prediction accuracy considering varying forecasting horizons (the number of days in advance when making the prediction) at three distinct end dates. These dates are selected based on the previous empirical results to be representatives of the different waves. This setting mirrors the real-life context where decisions are often needed to be made several days in advance.

The outcome, displayed in Figures 5(a), 5(b), and 5(c), shows an expected trend: an increased forecasting horizon generally corresponds to decreased prediction accuracy. This trend can be attributed to the increased challenges introduced by longer response times. However, there are instances where model accuracy improves with an increased forecasting horizon, likely due to the inherent variability in the data. Notably, on all three different dates, GPR and MLP models perform the best likely due to their smaller parameter count and simpler structure. Based on the results, we make the recommendation that 6 to 12 days is a good trade-off between a longer forecasting horizon and better prediction accuracy as the prediction error generally does not increase much during this period.

## 6 CONCLUSIONS

In this study, we explored the feasibility of utilizing publicly available wastewater data to forecast the number of COVID-19 cases. We employed five representative time-series prediction methods to capture the temporal associations within the viral wastewater concentration levels and case count data. Our empirical results show that the resulting models performed comparably with those trained on a more diverse range of data sources, underscoring the viability of this approach even with restricted data access.

Furthermore, our research underscores the importance of data segmentation during evaluation to better comprehend the inherent relationship between wastewater data and COVID-19 case count. This segmentation approach addresses the complexities posed by testing data spanning multiple waves, which can influence model evaluation metrics. Grounded in our empirical findings, we also propose practical guidelines regarding the forecasting horizon for case count prediction.

We hope that the findings of this study contribute to the growing body of research on wastewater-based epidemiology and provide valuable insights into the challenges and potential solutions for accurate epidemic forecasting using wastewater data, which can be applied in real-world scenarios to improve public health surveillance and inform decision-making processes. We acknowledge the complexities introduced by evolving testing and reporting practices during the COVID-19 pandemic, which make it increasingly hard to acquire ground truth data, and therefore alternative metrics like mortality data may gain prominence in different stages of epidemiological forecasting. We also acknowledge the existence of other publicly accessible data sources of varying types that may be utilized, including reproductive number[12], hospitalization numbers, and mortality rates[10, 36]. These additional data sources present ample opportunities for future research directions, broadening the scope of our current understanding and forecasting capabilities of public health scenarios.

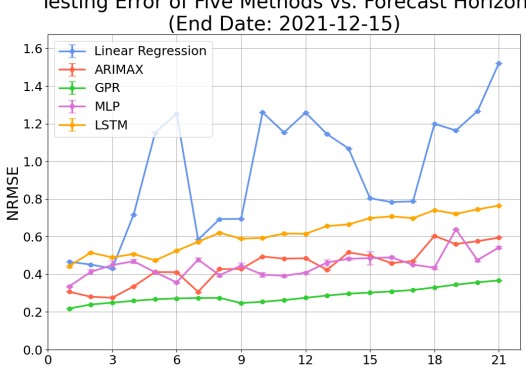

(a) Performance comparison w.r.t. #days to react on 2021-12-15

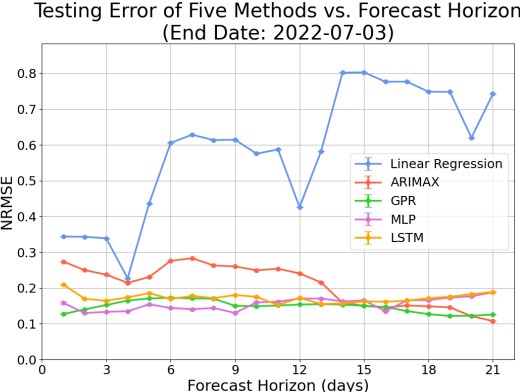

(b) Performance comparison w.r.t. #days to react on 2022-07-03

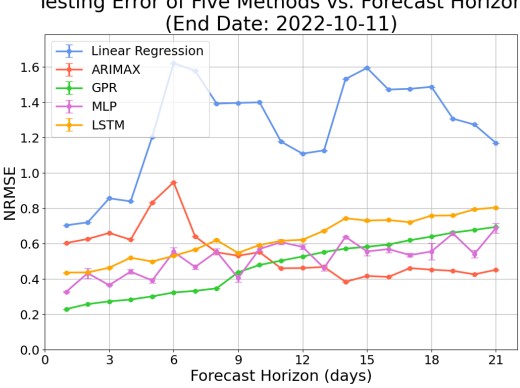

(c) Performance comparison w.r.t. #days to react on 2022-10-11

**Figure 5: Prediction accuracy corresponding to different lead times at three different dates. The x-axis indicates the forecasting horizon, and the y-axis denotes the normalized root mean square error (NRMSE) of the prediction of the number of cases. The three different dates are chosen to illustrate the models' performance at distinct waves during the pandemic.**

## ACKNOWLEDGMENTS

Zhicheng Zhang, Fei Fang, Angel Desai, and Sonja Neumeister were supported in part by grant SES2200228 from the National Science Foundation. Maimuna Shahnaz Majumder was supported in part by grant R35GM146974 from the National Institute of General Medical Sciences, National Institutes of Health. The funders had no role in study design, data collection and analysis, decision to publish, or preparation of the manuscript. Zhicheng Zhang was supported in part by SCS Dean's Fellowship.

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
