# OpenReview forum: "Unlocking the Potential of Public Datasets: Wastewater-Based Epidemiological Forecasting During COVID-19"
_KDD.org/2023/Workshop/epiDAMIK — KDD 2023 Workshop epiDAMIK_

### Official Review · Reviewer_qH4r · 2023-06-28
**The work is somewhat significant**

**Rating:** 2
**Confidence:** 3

**Review:**

## Clarity

This paper is easy to read however I found it hard to fully understand the proposed method

## Quality

The work is well-motivated but the benefits of proposed methods are not clear.

## Originality

Using ML methods for the two examined datasets is original.

## Significance

The work is somewhat significant.


## Pros:

- Have diverse methods for modeling the time-series data.

- The result for wave-based segmentation is interesting but needs more explanation.

- The experiment results show better performance compared with the previous method (Random Forest) with additional data sources, however, I am concerned about the evaluation process that is not consistent between the two works.

- Results show the potential of ML methods can get competitive results without additional data sources.

## Cons:

- The proposed method is poorly explained


- In equation (1).  $\hat{x}$ is not mentioned before, and the condition part authors compare $x_t$ with $T$ which is confusing.

- “AD and MSM.” abbreviations need an explanation.

- Need to introduce the role of $\alpha, \beta$ in getting the final change points

- The benefit of the technique dealing with misaligned time-series data is not clear.

- Authors try to deal with the distribution shift problem by applying wave-based segmentation on test data, however, segmentation removes the variety of the trend inside one segment then it seems to be easier for models to predict.

---

### Official Review · Reviewer_7Jp8 · 2023-06-29
**WBE Forecasting during Covid-19**

**Rating:** 3
**Confidence:** 4

**Review:**

# Clarity
I found the work easy to follow and well written.

# Quality
The applications of the work are clear for epidemiologists and data scientists.

# Originality
The data set used is novel, but the methods themselves are well studied and fairly straightforward.



The authors describe their experiments for using wastewater based epidemiology (WBE) methods for case count prediction at the national level versus traditional epidemiological methods, which may require more extensive and less commonly available data. The experiments show similar levels of accuracy for prediction of case counts moving forward given prior time series data.

The work is novel in the questions that it asks and the analysis that it provides on the foundations of WBE. I think that the data set itself could be expanded on, however. The specific value being measured against is only mentioned in Figure 1 (Effective copies of genome per $\mu L$ ), and it is unclear if there are other predictive factors being looked at.

The authors note that they aggregate wastewater data to a country level for making predictions due to the biases in data collection, but is national level data granular enough to be useful? The authors could do an analysis on the more regional data as well to see if the accuracy of their predictions holds up at the county/city level. This could be used as evidence for expansion of this data collection into these more rural areas as well.

---

### Official Review · Reviewer_5gRo · 2023-07-01
**Review of Unlocking the Potential of Public Datasets: Wastewater-Based Epidemiological Forecasting During COVID-19**

**Rating:** 4
**Confidence:** 2

**Review:**


Summary: Employing several different models, this paper demonstrates how aggregated wastewater data from across the US can be used to forecast COVID-19 cases. This paper also evaluates the optimal horizon for forecasting COVID-19 case data from wastewater signals.

Clarity:  This paper was well written and the paper’s objectives are clear. There are also clear descriptions for why given models were chosen for this evaluation.

To improve upon the clarity, I would suggest the following:

--Further explain why case counts were used instead of hospitalization counts as the COVID-19 outcome metric. The given explanation in the paper is that, “...case count data becomes an effective indicator of the strain on the healthcare system and the potential long-term effects of SARS-CoV-2 infection”. However, this same logic applies to COVID-19 hospitalization data, which did not suffer from the same notorious underreporting as case data did. This is not to say case data shouldn’t be used, just it is not clear why this was the outcome metric chosen.


-- it is unclear at what time stamp “ground truth” data was being pulled. Did the authors use case data as-of the date of model evaluation (i.e., potentially revised case data)? Or only case data available the week of wastewater data collection?


--Based on figures 5a - 5c, the error (measured in NRMSE) does not look much worse at 9 days vs. 6 days. It would help if the authors could clarify more objectively how the cutoff for 6-8 days was determined as the optimal horizon period.


Minor comments on clarity:

--A short sentence or two about what is being measured in the wastewater would be beneficial. What gene is being targeted / measured to determine COVID-19 concentrations?

--Because there is so much variability in wastewater data, it might be helpful to mention how the prediction intervals of models are impacted by the changes in the wastewater data.

--In section 4.1.1, definitions are needed for variables in equation 1. It is not immediately apparent what each different “x” represents. Additionally, in section 4.1.1, the authors mention an evaluation of data delays in section 5.5, however section 5.5 is about horizons, and not data delays.

--A note that the hyperlinks for footnotes 1 and 2 are broken.


Originality: There are similar articles that have compared modeling approaches on their ability to predict COVID-19 cases from wastewater data, however, this paper adds to the growing body of literature by using a segmentation approach on publicly available data, as well as determining the ideal forecast based on prediction accuracy and maximizing a longer forecasting horizon.

Significance: The significance of this paper is that it demonstrates that numerous modeling approaches provide similar results when using wastewater data to predict COVID-19 cases at a national level. This paper could be improved by adding discussion of the biases and limitations of using data aggregated at a national level, and demonstrating how well these models perform at a smaller geographic scale. Discussion of the confidence levels of these models would also be beneficial, as would using multiple metrics to evaluate model performance.

Pros:

--Well written paper with many clear discussions about decisions made in the experimental process.

-- Use of publicly available data makes methods replicable.

-- Project opens the door for additional analyses that can be done using wastewater data, as well as additional variables that can be added to the analysis.

Cons:

--As noted in sections above, the paper could expand on the limitations of using wastewater and case data at the national level, such as heterogeneity in COVID-19 across the county, non-standardized collection approaches across counties / states, variation in sampling sites, etc.


--The determination of why 6-8 days is the optimal horizon is not clear from the figures presented.